# Therapeutic Potential of Stearoyl-CoA Desaturase1 (SCD1) in Modulating the Effects of Fatty Acids on Osteoporosis

**DOI:** 10.3390/cells13211781

**Published:** 2024-10-28

**Authors:** Young-Jin Seo, Jin-Ho Park, June-Ho Byun

**Affiliations:** 1Department of Oral and Maxillofacial Surgery, Gyeongsang National University School of Medicine and Gyeongsang National University Hospital, Institute of Medical Sciences, Gyeongsang National University, Jinju 52727, Republic of Korea; seoyoung1126@gnu.ac.kr; 2Department of Convergence Medical Science, Gyeongsang National University, Jinju 52727, Republic of Korea; 3Department of Nutritional Science, University of Michigan School of Public Health, Ann Arbor, MI 48109, USA; jinhop@umich.edu

**Keywords:** osteoporosis, MAT, BM-MSCs, stearoyl-CoA desaturase 1, monounsaturated fatty acids

## Abstract

Osteoporosis is a common skeletal disease, primarily associated with aging, that results from decreased bone density and bone volume. This reduction significantly increases the risk of fractures in osteoporosis patients compared to individuals with normal bone density. Additionally, the bone regeneration process in these patients is slow, making complete healing difficult. Along with the decline in bone volume and density, osteoporosis is characterized by an increase in marrow adipose tissue (MAT), which is fat within the bone. In this altered bone microenvironment, osteoblasts are influenced by various factors secreted by adipocytes. Notably, saturated fatty acids promote osteoclast activity, inhibit osteoblast differentiation, and induce apoptosis, further reducing osteoblast formation. In contrast, monounsaturated fatty acids inhibit osteoclast formation and mitigate the apoptosis caused by saturated fatty acids. Leveraging these properties, we aimed to investigate the effects of overexpressing stearoyl-CoA desaturase 1 (SCD1), an enzyme that converts saturated fatty acids into monounsaturated fatty acids, on osteogenic differentiation and bone regeneration in both in vivo and in vitro models. Through this novel approach, we seek to develop a stem cell-based therapeutic strategy that harnesses SCD1 to improve bone regeneration in the adipocyte-rich osteoporotic environment.

## 1. Introduction

Human bones undergo a continuous process of renewal through bone remodeling, where osteoclasts resorb old bone while osteoblasts form new bone [1,2]. However, with aging, bone resorption often outpaces bone formation, leading to an imbalance that results in osteoporosis, characterized by decreased bone density and increased susceptibility to fractures [3,4]. Osteoporosis can be categorized as primary or secondary. Primary osteoporosis is commonly associated with aging, genetic factors, and estrogen deficiency during menopause, accounting for the majority of cases. In contrast, secondary osteoporosis can be caused by specific diseases or medications, even affecting younger individuals. Given its multifactorial nature, osteoporosis is influenced by a variety of conditions and treatments [5,6,7,8,9,10].

Current therapies for osteoporosis, such as bisphosphonates, anti-resorptive agents, and estrogen modulators, primarily target the reduction of bone resorption [11,12,13]. However, these treatments do not promote bone regeneration or offer permanent solutions. In this study, we utilize ovariectomized (OVX) rats to model post-menopausal osteoporosis induced by estrogen deficiency [14]. Estrogen deficiency promotes the differentiation of monocytes into osteoclasts, increasing bone resorption while inhibiting the differentiation of mesenchymal stem cells (MSCs) into osteoblasts and promoting their differentiation into adipocytes [15,16]. This shift is further compounded by reduced levels of PGC-1α, elevated SOST, and increased follicle-stimulating hormone (FSH), all contributing to increased adipogenesis and decreased bone turnover. Consequently, the bone marrow undergoes a rise in marrow adipose tissue (MAT), which is also seen in conditions such as caloric restriction, high-calorie diets, and aging-related osteoporosis [17].

MAT plays a unique role in both systemic and skeletal metabolism, differing from white and brown adipose tissue in function and composition. MAT can be classified into two main forms: constitutive marrow adipose tissue (cMAT) and regulated marrow adipose tissue (rMAT) [18]. cMAT forms early in life and is relatively stable, whereas rMAT emerges later in life, fluctuates with age and disease, and contains higher levels of saturated fatty acids (SFAs). In conditions such as estrogen deficiency, rMAT, which is rich in SFAs, increases significantly. This is particularly relevant in osteoporosis models, where rMAT’s presence alters the bone microenvironment by influencing fatty acid metabolism and energy usage by osteoblasts [19].

SFAs, such as palmitic acid (PA) and stearic acid (SA), have been shown to reduce osteoblast activity and negatively correlate with bone mineral density (BMD) [20]. They impair osteoblast function by reducing β-catenin levels and inhibiting key osteogenic transcription factors like Runx2. Moreover, SFAs increase inflammation through TLR4 signaling and promote lipotoxicity by inducing reactive oxygen species (ROS), leading to apoptosis [21,22,23,24]. Conversely, monounsaturated fatty acids (MUFAs), such as oleic acid (OA), have been found to mitigate these effects, reducing SFA-induced apoptosis, promoting angiogenesis, and enhancing bone regeneration. Importantly, MUFAs also suppress the inflammatory effects of TNF-α and support osteoblast differentiation [25,26,27].

Given these findings, this study aims to explore the potential therapeutic effects of converting SFAs to MUFAs in an osteoporotic environment characterized by increased MAT. Specifically, we focus on the enzyme stearoyl-CoA desaturase 1 (SCD1), which catalyzes the conversion of stearic acid (C18:0) and palmitic acid (C16:0) to oleic acid (C18:1) and palmitoleic acid (C16:1), respectively [28,29,30]. By overexpressing SCD1 in MSCs, we seek to counteract the negative effects of SFAs on osteoblast differentiation and bone regeneration, providing a novel approach to enhance osteogenesis and reduce osteoclastogenesis in osteoporosis (Figure 1).

## 2. Materials and Methods

### 2.1. In Vivo Study

#### 2.1.1. Animal Modeling (Osteoporosis Rat)

All animal procedures were performed by the Institutional Animal Care and Use Committee (IACUC) protocol (GNU-231017-R0194) of Gyeongsang National University. The animals were provided with free access to food and housed in a well-ventilated, clean animal room with a 12 h light/dark cycle. Eight-week-old female Sprague-Dawley rats were randomly assigned to either the sham-operated group (Sham) or the ovariectomized (OVX) group, with 5 rats in each group. Anesthesia was induced using a mixture of 2.0–3.0% isoflurane and oxygen. Bilateral ovariectomy was performed on the rats in the OVX group, with careful ligation of the oviduct. In the Sham group, rats underwent a sham surgery in which the same amount of adipose tissue surrounding the ovary was removed. The animals were housed in the facility for 8 weeks postoperatively [14], a period required for the development of osteoporosis. Body weight was monitored weekly, and bone mineral density (BMD) was measured using Dual Energy X-ray Absorptiometry (DEXA) (OsteoSys, Seoul, Korea) to evaluate the progression of osteoporosis. All DEXA images and quantitative data were analyzed using Insight software (Version 1.0.6; OsteoSys). The region of interest (ROI) for DEXA was set at the distal femur.

#### 2.1.2. Animal Modeling (Bone Defect Rat)

A bone defect was created in the femur of the OVX rat model. After shaving and disinfecting the skin, a 1.5 cm incision was made on the lateral side of the knee to expose the muscle. Blunt dissection of the muscles was performed to expose the femoral condyle while minimizing damage to muscles, vessels, and nerves. A bone defect, approximately 3 mm in diameter, was drilled into the femur using a round bur. During drilling, saline was applied to cool the area and flush out bone fragments. The rats, including those in the Sham and OVX groups, were implanted with either Ad-CTL or Ad-SCD1-treated BM-MSCs combined with fibrin glue, or they were assigned to the negative control (N.C) group, which received no cells. The cells were treated with Ad-CTL or Ad-SCD1 virus at a multiplicity of infection (MOI) of 20. Then, the cells (2.5 × 10^6^ cells/mL) were mixed with fibrinogen from Greenplast Q (GC Pharma, South Korea). The cell suspension with Component A was then inserted into a Greenplast Q syringe and mixed with thrombin at a 1:1 (vol/vol) ratio for implantation into the rat femur defect model. The periosteum and skin were sutured with 4–0 Vicryl sutures (Ethicon). Bone mineral density (BMD) of the femur defect area was measured using DEXA to evaluate defect healing at 1, 3, 5, and 7 weeks post-surgery. All DEXA images and quantitative data were analyzed using Insight software 9.64. The region of interest (ROI) was set specifically on the femur defect.

#### 2.1.3. μCT Analysis

After euthanasia, the femurs were removed and stored 4% paraformaldehyde for 24 h via μ-CT (Skyscan 1172, Bruker, Kontich, Belgium). The acquired CT images were 3D reconstructed with NRecon (Bruker) or a 3D visualization program (CTVox, Bruker) based on the acquired CT images, respectively. The morphometric parameters, namely, bone volume/total volume (BV/TV), are measured at a growth plate 2 mm toward the diaphysis83. Osteoporosis was confirmed by uCT and morphometric analyses of the femurs from OVX and Sham rats.

#### 2.1.4. Immunochemistry

The femur samples were decalcified in RDO Gold (Apex Engineering Products Corporation, Aurora, IL, USA) for 7 days, embedded in paraffin, and sectioned into 5 μm thick slices. The sections were stained with hematoxylin and eosin (H&E) or used for immunohistochemistry (IHC) and observed under a light microscope. IHC analysis for CD36 was performed on the decalcified, paraffin-embedded biopsy specimens. All relevant reagents were purchased from Vector Laboratories (Burlingame, CA, USA). The sections were deparaffinized, rehydrated, and subjected to antigen retrieval for 5 min at 80 °C. They were then blocked with 2.5% normal horse serum for 20 min at room temperature (RT). After blocking, the sections were incubated with a primary antibody against CD36 (#28109, Cell Signaling Technology, Beverly, MA, USA) at a 1:400 dilution for 30 min at RT. Following the primary antibody incubation, the sections were incubated with secondary biotinylated antibodies for 30 min at RT. The sections were then washed and incubated with an avidin–biotin complex for 30 min at RT, followed by washing and development using a 3,3′-diaminobenzidine (DAB) peroxidase substrate. After development, the sections were dehydrated, cleared in xylene, and mounted.

### 2.2. Characterization of BM-MSCs

Bone marrow was isolated 8 weeks post-OVX surgery from 16-week-old Sham and OVX female Sprague Dawley rats (n = 5 per group). The femurs were removed, the epiphyses were excised, and the medullary canals were flushed to collect bone marrow cells. The cells were cultured in a growth medium consisting of the alpha minimum essential medium, 20% fetal bovine serum, and 1% penicillin/streptomycin until they reached 80% confluence. The cultures were maintained at 37 °C in 95% humidified air with 5% CO. To confirm the presence of mesenchymal stem cells (MSCs) through flow cytometry, BM-MSCs were divided into tubes, with 1 × 10^6^ cells per tube. The following dye-labeled antibodies were added to each tube: anti-rat CD29 APC antibody and anti-rat CD90 APC for MSC markers, and anti-rat CD34-FITC and anti-rat CD45 FITC for hematopoietic markers. The tubes were gently mixed and incubated for 30 min at 4 °C, then washed three times with FACS buffer to remove unbound antibodies. Flow cytometry was performed using a BD LSR Fortessa Flow Cytometer (Becton-Dickinson, San Jose, CA, USA), and data were processed using FACS Diva software v8.0.1 (Becton-Dickinson).

### 2.3. Evaluation of the Osteogenic and Adipogenic Differentiation

BM-MSCs were cultured at passages 4–8 in osteogenic induction medium, which consisted of α-MEM supplemented with 10% FBS, 1% penicillin/streptomycin, 50 µg/mL L-ascorbic acid 2-phosphate, 10 nM dexamethasone, and 10 mM β-glycerophosphate at a density of 5 × 10^4^ cells/well in a 24-well co-culture plate for 3 weeks [31,32]. The medium was changed every 3 days. Adipogenic differentiation was induced in confluent BM-MSCs by incubation in α-MEM containing L-glutamine, 10% FBS, 1% penicillin/streptomycin, 1 µM dexamethasone, 0.5 mM indomethacin, 50 µM 3-isobutyl-1-methylxanthine, and 10 µg/mL insulin, at a density of 3 × 10^4^ cells/insert in an indirect co-culture plate for 3 weeks. The medium was changed every 3 days during adipogenic differentiation.

To evaluate the effects of rat BM-MSCs on osteogenic differentiation of osteoblasts and adipocytes, all media were changed every 2–3 days. For the calcium deposition experiment, osteoblasts were decalcified in RDO Gold for 24 h at room temperature. Calcium concentration in the supernatant was assessed using a calcium assay kit (Abcam, Waltham, MA, USA) via spectrophotometry. Histochemical analysis of alkaline phosphatase (ALP) and alizarin red S (ARS) staining was performed using NBT/BCIP substrate solution (Pierce Chemical Co., Dallas, TX, USA) or a 2% ARS solution. ALP activity was measured using a TRACP and ALP assay kit (Takara Bio Inc., Shiga, Japan) according to the manufacturer’s instructions [31,32]. Adipocyte differentiation was evaluated by Oil Red O (ORO) staining to identify cytoplasmic lipids in periosteal cells after 14 days of culture. Formaldehyde-fixed cells were stained with 0.5% Oil Red O (*w*/*v*) (Sigma Aldrich, St Louis, MO, USA) in 60% isopropanol, washed with distilled water, and visualized under light microscopy.

### 2.4. Construction of Recombinant Adenovirus Expressing Human-SCD1

The plasmid containing the cDNA of human SCD1 (NM_005063) (pCMV6-XL5-SCD1) was purchased from Origene. Using pCMV6-XL5-SCD1 as a template, fragments of the SCD1 gene were amplified with primers capped with XhoI and HindIII restriction sites. The forward primer was 5′ CTC GAG CCA CCA TGC CGG CCC ACT TGC TG 3′, and the reverse primer was 5′ AAG CTT TCA GCC ACT CTT GTA GTT TCC ATC T 3′. This fragment was ligated into the pGEM-T Easy vector (Promega, Fitchburg, WI, USA) for amplification, then digested with XhoI and HindIII restriction enzymes and subcloned into the corresponding sites of the pAd-Track-CMV vector. Recombinant adenoviruses were constructed using the Ad-Easy-1 system, where the adenoviral construct was generated in BJ-5183 bacterial cells [33,34]. Correct clones were propagated in RecA DH5α cells. The recombinant adenoviral vectors were linearized with PacI and used to infect 293A cell cultures. The recombinant adenoviruses were then amplified and titrated by plaque assay, typically yielding titers of approximately 2.0 × 10^8^ plaque-forming units (pfu)/mL.

For transduction, BM-MSCs were grown in α-MEM containing 10% FBS and 1% penicillin/streptomycin for 2 days. The cells were infected with Ad-SCD1 or a control adenovirus expressing GFP (Ad-CTL) at a multiplicity of infection (MOI) of 20 in α-MEM supplemented with 2% FBS and 1% penicillin/streptomycin. Four hours post-infection, the medium was replaced with fresh α-MEM containing 10% FBS. The viability of the BM-MSCs was assessed using a CCK-8 cell counting kit (Dojindo Molecular Technologies, Rockville, MD, USA).

### 2.5. Western Blot Analysis

BM-MSCs were lysed in RIPA buffer (Cell Signaling Technology, Beverly, MA, USA), and protein extraction was performed on ice. Protein concentration was determined using a BCA assay. Protein lysates were then mixed with SDS loading buffer, and proteins were separated by electrophoresis on a 10% SDS-PAGE gel. Subsequently, proteins were transferred onto polyvinylidene difluoride (PVDF) membranes (Millipore, Burlington, MA, USA). After blocking with 5% bovine serum albumin (BSA) in TBST (Tris-buffered saline with Tween 20) for 1 h, the membranes were washed and incubated sequentially with a primary anti-SCD1 antibody (#2794, Cell Signaling Technology, Beverly, MA, USA) and anti-rabbit horseradish peroxidase (HRP)-conjugated secondary antibodies. Specific antibody binding was visualized using an enhanced chemiluminescence (ECL) detection reagent (Invitrogen, Carlsbad, CA, USA).

### 2.6. Statistical Analysis

Each experiment was performed independently, at least three times. The results of one of the three independent experiments are shown as representative data. Data are expressed in mean standard deviation. Statistical analyses were computed using GraphPad Prism 9 (GraphPad Software 8.2.1, San Diego, CA, USA). Tests used include one-way ANOVA, two-way ANOVA, or two-tailed Student’s *t*-test. Comparisons with *p* < 0.05 were considered significant.

## 3. Results

### 3.1. Evaluation of Osteoporosis Modeling in Rats

At 8 and 16 weeks post-ovariectomy surgery (Figure 2A), micro-CT imaging was used to assess the distal femurs. Osteoporosis severity was evaluated by measuring a 2 mm bone volume in the proximal direction from the growth plate of the rat femur. In the OVX group, the femoral trabeculae appeared sparse, and the cortical bone showed thinning following ovariectomy (Figure 2B). Comparative analysis with the Sham group revealed significantly lower values of bone mineral density (BMD) and bone volume fraction (BV/TV) in the OVX group, while body weight showed a significant increase in the OVX group starting at 2 weeks post-surgery (Figure 2C–E). These findings confirm the successful establishment of the osteoporosis model, with marked deterioration of bone microstructure evident at 8 and 16 weeks post-OVX surgery. Histological analysis demonstrated an increase in marrow adipose tissue (MAT) following OVX (Figure 2F,G). CD36 immunoreactivity (brown) was observed in distal femur biopsy specimens from both Sham and OVX rats 16 weeks after surgery. CD36-positive osteoblasts (OBs) were present on osteoid surfaces (OSs) within the trabecular bone of the OVX group, whereas CD36-negative osteoblasts were observed in the Sham group. Both the Sham and OVX groups exhibited CD36-positive vascular structures, though adipocytes within the marrow cavity showed higher CD36 positivity in the OVX group compared to the Sham group. Additionally, faint staining of CD36 was detected in osteoclasts (OCs) located on eroded surfaces (ESs) in the OVX group (Figure 2H). These results suggest that fatty acid uptake was increased in the distal femur and, more specifically, that the distal cancellous bone was significantly affected by fatty acids. This motivated the designation of the distal femur as a target site for bone defects.

### 3.2. Characterization of BM-MSCs

At 10 days of cell culture, both Sham and OVX BM-MSCs exhibited substantial expression of the MSC surface markers CD29 and CD90, confirming their MSC characteristics (Figure 3A). In contrast, the expression of hematopoietic markers CD34 and CD45 was very minimal. These surface marker evaluations confirmed that the majority of both cell populations consisted of MSCs. Osteoblast differentiation was further verified through alkaline phosphatase (ALP) staining and alizarin red S (ARS) staining, while adipocyte differentiation was confirmed by Oil Red O (ORO) staining, providing additional confirmation of MSC characterization (Figure 3B). There was no significant difference in the degree of calcium deposition or lipid droplet formation between the OVX and Sham groups. However, the expression of ALP, an early marker of osteoblast differentiation, was significantly reduced in the OVX group.

### 3.3. Effect of Adipocytes on Osteogenesis by Co-Culture

To investigate the effect of adipocytes on osteoblast differentiation, a co-culture system was established (Figure 4A). ALP and ARS staining revealed that osteoblast differentiation was reduced due to the influence of adipocytes in both Sham and OVX conditions. Notably, OVX BM-MSCs exhibited weaker ALP expression than the control group, even in the absence of adipocytes. Additionally, adipocytes significantly reduced calcium deposition in the OVX group compared to the control group (Figure 4B). These findings suggest that the osteogenic differentiation potential of BM-MSCs is further suppressed in the osteoporotic microenvironment of OVX, which is characterized by increased marrow adipose tissue (MAT). In this study, a palmitic acid treatment experiment was performed to confirm the effect of saturated fatty acids, which are assumed to be one of the main factors of this phenomenon, on osteoblast differentiation. As a result of treating at various concentrations within a range that did not affect cell viability, it was observed that the degree of osteoblast differentiation was reduced in a concentration-dependent manner (Figure 4C–E). This suggests the possibility that saturated fatty acids may be one of the main causes of decreased osteoblast differentiation due to adipocytes.

### 3.4. Effect of SDC1 Overexpression on PA-Induced Decrease in Osteo-Differentiation

In response to the decreased osteogenesis of BM-MSCs following palmitic acid (PA) treatment, we aimed to mitigate this effect by overexpressing stearoyl-CoA desaturase 1 (SCD1). Using an adenovirus-based approach, we generated Ad-SCD1 for SCD1 overexpression (Figure 5A). A multiplicity of infection (MOI) range was identified that did not impact cell viability, ruling out any detrimental effects caused by the virus itself (Figure 5B). Western blot analysis confirmed SCD1 overexpression in a dose-dependent manner (Figure 5C). Successful viral transduction into cells was further validated by GFP expression (Figure 5D). Additionally, we reconfirmed that both Ad-SCD1 and the control adenovirus (Ad-CTL) had no adverse effects on BM-MSC viability (Figure 5E). Importantly, the osteoblast differentiation potential of BM-MSCs, which had been diminished by PA treatment, was significantly restored following the introduction of Ad-SCD1, as evidenced by increased ALP activity and calcium deposition levels (Figure 5F–I). These findings suggest that SCD1 holds promise for enhancing osteoblast differentiation in environments rich in saturated fatty acids, such as those induced by PA.

### 3.5. In Vivo Study: The Effects of SCD1 on Bone Regeneration in the Osteoporosis Model

To further explore SCD1’s potential to enhance osteoblast differentiation and promote bone regeneration, we investigated its effects in an in vivo osteoporosis model characterized by expanded marrow adipose tissue (MAT) and increased fatty acid (FA) resorption. In this experiment, a 3 mm diameter bone defect was created in the distal femur of rats with OVX-induced osteoporosis. BM-MSCs transduced with either Ad-CTL or Ad-SCD1 were transplanted using fibrin glue (Figure 6A). Negative control groups included Sham and OVX models without cell transplantation. The study was conducted over 8 weeks to evaluate bone regeneration, with bone mineral density (BMD) measurements performed at 1, 3, 5, and 7 weeks using DEXA (Figure 6B). Up until week 3, there were no significant differences among the four groups, but notable differences began to emerge by 5 weeks. Interestingly, the group transplanted with SCD1-overexpressing cells in the osteoporosis model showed bone regeneration capacity similar to the non-transplanted Sham model. In contrast, the bone regeneration capacity of the group transplanted with control virus-treated cells and the non-transplanted OVX group was comparable (Figure 6C). These results suggest that SCD1 plays a positive role in bone defect recovery in the osteoporosis model. This experiment indicates that SCD1 may help convert saturated fatty acids in adipocytes, which are elevated in osteoporosis, into unsaturated fatty acids, thereby enhancing osteoblast activity and ultimately improving bone regeneration capacity.

## 4. Discussion

The increase in marrow adipose tissue (MAT) in osteoporosis models has been well documented in numerous studies, including estrogen-deficiency models like the ovariectomized (OVX) rat, which is a widely recognized post-menopausal osteoporosis model [15,35,36,37]. Notably, the rise in MAT, particularly in saturated fatty acid (SFA)-rich MAT, has been linked to alterations in the bone microenvironment [18,35,38]. Extensive research has focused on the effects of this MAT increase on bone, exploring the roles of various adipokines and their influence on bone cells [39,40,41].

In this study, the increase in MAT and fatty acid (FA) uptake by osteoblasts evidenced by CD36 upregulation represents a distinct phenomenon in the osteoporotic bone microenvironment [40,41]. This surge in MAT and enhanced FA uptake highlights the need for a novel approach in therapeutic strategies, as conventional mesenchymal stem cell (MSC) therapies designed for a healthy bone environment may not be suitable for osteoporotic conditions. Despite the well-established link between MAT and osteoporosis, relatively few studies have investigated the role of MAT in MSC-based therapies for osteoporosis patients [42,43]. Given that SFAs such as palmitic acid (PA) and stearic acid (SA) are known to reduce the proliferation and osteogenesis of MSCs and osteoblasts, we aimed to counteract this negative effect by overexpressing stearoyl-CoA desaturase 1 (SCD1), an enzyme that converts SFAs into monounsaturated fatty acids (MUFAs), such as palmitoleic and oleic acids. Fatty acid (SFA and MUFA) composition plays critical roles in modulating the fate and function of bone cells. SFAs impair osteoblast differentiation by reducing key markers (Runx2, β-catenin) and promoting inflammation through TLR4 activation, leading to increased ROS production, apoptosis, and impaired bone formation. SFAs also enhance osteoclast differentiation by increasing pro-inflammatory cytokines (TNF-α, IL-1β). In contrast, MUFAs suppress inflammation, promote osteoblast differentiation, and reduce apoptosis. MUFAs also inhibit osteoclastogenesis by downregulating RANKL and increasing osteoprotegerin (OPG) expression, improving bone formation [22,23,24,25,26,27,44]. To test whether the reduced osteogenesis in an SFA-rich environment, such as one containing PA, could be reversed by SCD1 overexpression in rat BM-MSCs, we constructed an adenoviral vector to overexpress SCD1 (Ad-SCD1). Previous studies utilized liver X receptor (LXR) agonists to increase SCD1 expression, but we explored the direct overexpression of SCD1 itself using Ad-SCD1 [28]. Our results demonstrate that Ad-SCD1 effectively restored osteogenesis, as evidenced by increased alkaline phosphatase (ALP) activity and alizarin red S (ARS) staining, suggesting that SCD1 can mitigate the inhibitory effects of SFAs by converting them into MUFAs.

Additionally, we investigated whether MSC therapy, combined with SCD1 overexpression, could enhance bone regeneration in an OVX rat model with increased MAT. We targeted the distal femur bone defect, which is influenced by MAT, and found that bone mineral density (BMD) in the Ad-SCD1 group was significantly higher compared to other groups, suggesting that SCD1 contributes to improved bone healing by mitigating the negative effects of SFAs on osteogenesis in the OVX model. However, several limitations of this study need to be addressed. First, further research is required to confirm whether SFAs reduce proliferation through ROS-induced apoptosis and decrease osteogenesis by downregulating Runx2 expression [21,22,23]. Second, gas chromatography should be used to directly confirm changes in FA composition and the activity of SCD1. Third, increasing the number of animals per group in future in vivo experiments is essential for more robust statistical analysis, and additional assessments, such as micro-CT, should be performed to better quantify bone regeneration.

Despite these limitations, our study presents a novel therapeutic approach to addressing osteoporosis. Current osteoporosis treatments, such as bisphosphonates, antiresorptive agents, and estrogen modulators, primarily target the reduction of bone resorption but often fall short in promoting bone regeneration or providing a long-term solution. In contrast, our research highlights the role of increased marrow adipose tissue (MAT), particularly in patients with obesity or lean obesity, which is common in osteoporosis. Since the effects of coexisting fat cells persist even with ongoing treatments that reduce bone resorption, targeting the fatty acid composition within the bone microenvironment offers a promising new avenue for therapy. We believe that by converting harmful saturated fatty acids to beneficial monounsaturated fatty acids, as demonstrated by SCD1 overexpression, this approach could provide a foundation for future osteoporosis treatments that not only prevent bone loss but also promote bone regeneration.

## 5. Conclusions

In conclusion, this study highlights the significant influence of adipocytes in the progression of osteoporosis and proposes a novel therapeutic strategy to mitigate the disease by targeting this interaction. Using an osteoporosis rat model, we isolated BM-MSCs and conducted differentiation experiments into both osteoblasts and adipocytes. Our findings confirme that saturated fatty acids derived from adipocytes inhibit osteoblast differentiation. However, by overexpressing the SCD1 gene to convert these saturated fatty acids into unsaturated fatty acids, we demonstrated a promising approach to enhance bone regeneration. This strategy opens new avenues for more effective treatments and potentially overcoming osteoporosis.

## Figures and Tables

**Figure 1 cells-13-01781-f001:**
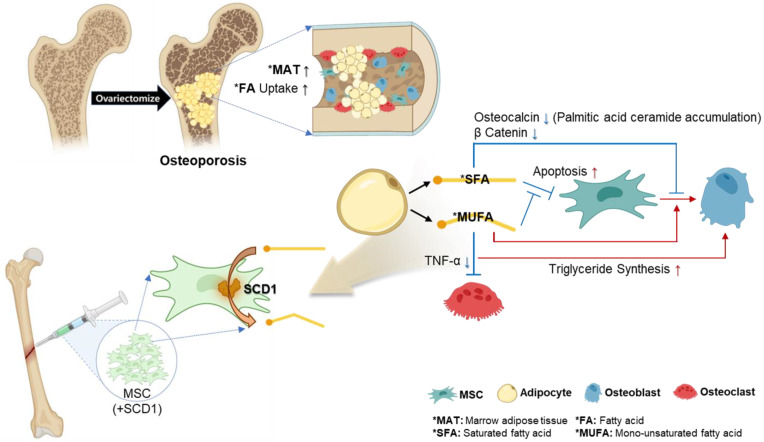
Schematic diagram illustrating the research concept. In osteoporosis models, fatty acids such as SFA and MUFA are secreted from expanding MAT. This study suggests that SCD1 can promote bone regeneration by converting SFA, which is known to have a negative effect on osteoblast differentiation and MSC proliferation, into MUFA, which can have a positive effect on bone metabolism. This highlights the therapeutic potential of MSCs through SCD1 overexpression in bone regeneration in osteoporosis models.

**Figure 2 cells-13-01781-f002:**
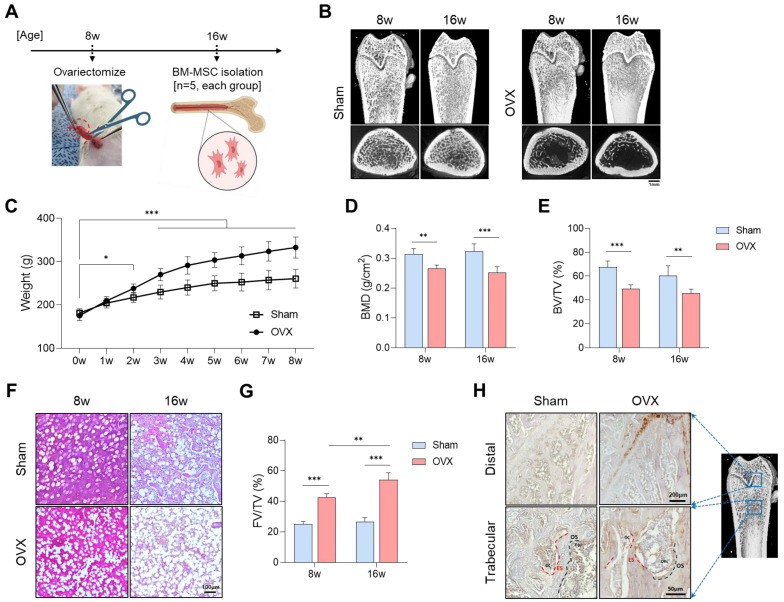
Evaluation of osteoporosis modeling in rats. (**A**) Schematic diagram of OVX surgery. (**B**) μCT images of femurs in the Sham and OVX groups. (**C**) Body weight (g). (**D**) Bone mineral density (BMD, g/cm^2^). (**E**) Bone volume fraction per total volume (BV/TV, %). (**F**) Histological examination of femoral bone marrow using H&E staining. (**G**) Quantification of fat volume fraction per total volume (FV/TV, %). (**H**) Immunostaining for CD36 in the distal femur and cancellous bone. (* *p* < 0.05, ** *p* < 0.01, *** *p* < 0.001).

**Figure 3 cells-13-01781-f003:**
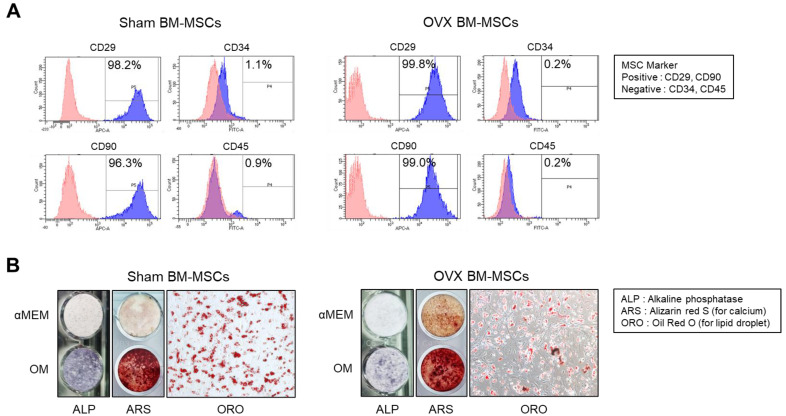
Characterization of BM-MSCs. (**A**) CD marker analysis by flow cytometry; MSC markers (CD29 and CD90), hematopoietic cell markers (CD34 and CD45). (**B**) Evaluation of osteoblast differentiation by ALP staining and ARS staining, and evaluation of adipocyte differentiation by ORO staining.

**Figure 4 cells-13-01781-f004:**
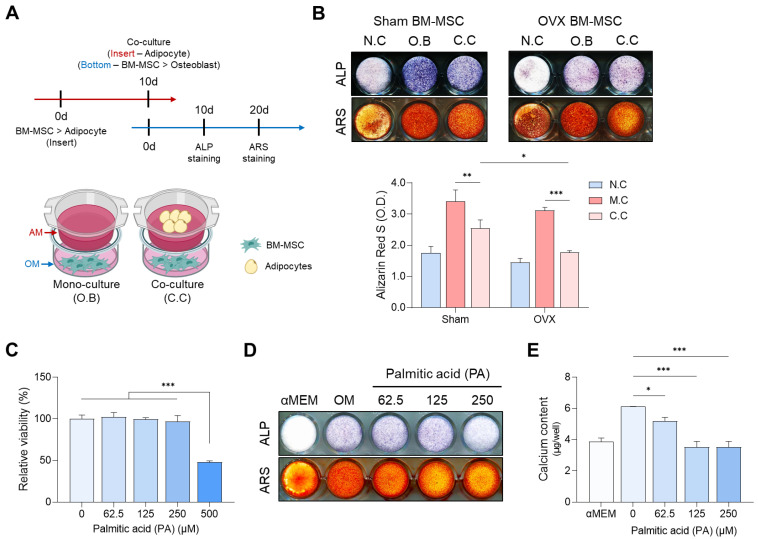
Effect of adipocytes on osteoblast differentiation. (**A**) Schematic representation of the co-culture experiment. (**B**) ALP and ARS staining on days 10 and 20 after co-culture. (**C**) Evaluation of cell viability at different concentrations of palmitic acid. (**D**) Evaluation of palmitic acid concentration-dependent osteogenic differentiation by ALP and ARS staining. (**E**) Evaluation of the degree of osteogenic differentiation by calcium content measurement. (* *p* < 0.05, ** *p* < 0.01, *** *p* < 0.001).

**Figure 5 cells-13-01781-f005:**
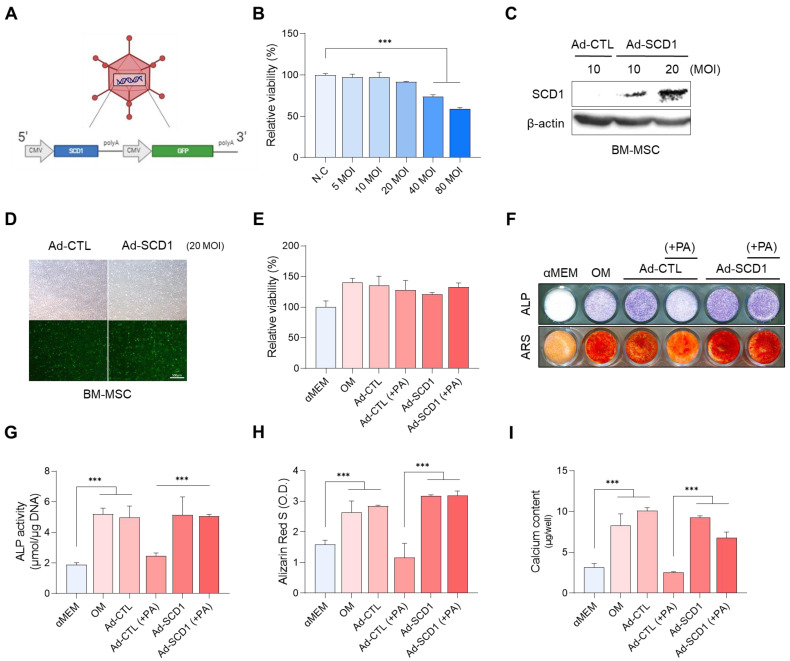
Effect of SCD1 overexpression on osteogenic differentiation reduced by palmitic acid (PA). (**A**) Production of adenovirus-based SCD1 vector. (**B**) Confirmation of cell viability of BM-MSCs according to virus MOI. (**C**) Verification of MOI-dependent overexpression of Ad-SCD1. (**D**) Confirmation of normal virus infection and GFP expression in BM-MSCs. (**E**) Measurement of cell viability with the virus. (**F**–**I**) Confirmation of the effect of Ad-SCD1 on osteogenic differentiation of PA-treated BM-MSCs through comparison of ALP and ARS staining, ALP activity, and calcium deposition. (*** *p* < 0.001).

**Figure 6 cells-13-01781-f006:**
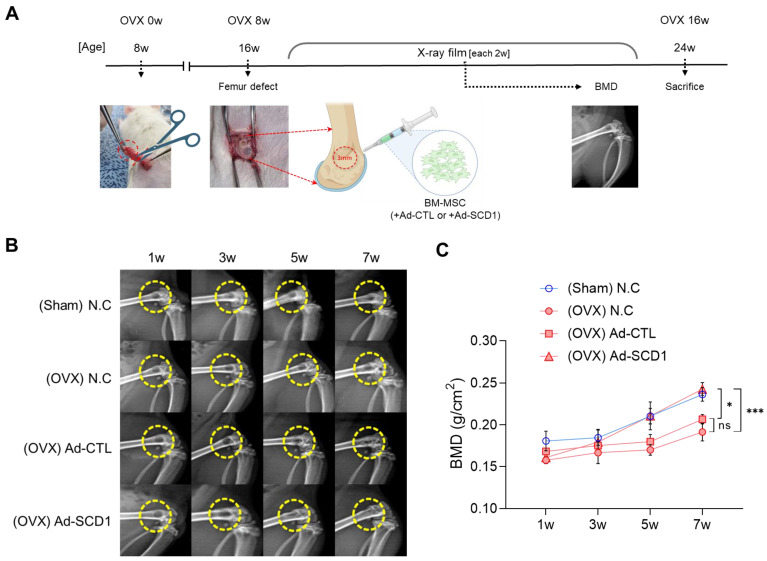
In vivo experiments to verify the effect of SCD1 on bone defect recovery in an osteoporosis model. (**A**) Schematic diagram of the animal modeling experiment. (**B**) X-ray scan images of the left distal femur at 1, 3, 5, and 7 weeks after cell transplantation into the defect. (**C**) BMD measurements by DEXA imaging at 1, 3, 5, and 7 weeks (* *p* < 0.05, *** *p* < 0.001).

## Data Availability

The datasets generated during and/or analyzed during the current study are available from the corresponding author upon reasonable request.

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
