# Peer review of "Therapeutic Potential of Stearoyl-CoA Desaturase1 (SCD1) in Modulating the Effects of Fatty Acids on Osteoporosis"

_cells, 2024, doi:10.3390/cells13211781_

Round 1

Reviewer 1 Report

Comments and Suggestions for Authors

In the current study, the authors presented the results of a series of experiments to demonstrate the effect of the enzyme SCD1 on modulating the altered osteoblastogenesis in an ovariectomized animal model. There are some minor errors to be corrected, and a few suggestions for presenting the results and discussion:

1. Line 95(p.3): it is believed that the rats used in the current study should be female; please correct the gender of the rats.

2. Lines 101-2(p.3): how do the authors ascertain that 8 weeks after the surgery is required to develop osteoporosis? If it is based on previous experiments, please indicate the reference.

3. Lines 222-4(p.6): it is difficult to tell if the authors use repeated measure ANOVA, a statistical method commonly used to analyze repeated measurements for humans or animals at different time points (such as the body weights of the rats in the current study). 

4. Line 345(p.11, Figure 6): it should be panel "C" that displays the BMD measurements by DXA (rather than B). Please correct.

5. The authors intended to examine whether the overexpression of the enzyme SCD1 can modulate the loss of osteoblast differentiation induced by estrogen deficiency. Though the results showed that the authors did successfully mitigate the adiposity in the bone marrow and increase the osteogenic differentiation in their model, it is pitful that the results did not show how the overexpression of SCD1 enzymes affects the fatty acid composition of bone marrow.  It is known that fatty acids in bone marrow microenvironment also affect the differentiation and behaviors of both osteoblasts and osteoclasts.  The authors may, at least, include in their discussion how fatty acid composition in bone marrow may influence the differentiation and/or the fate of both bone cells.

6. Similarly, the expression of cytokines and adipokines also varies with the proportion of saturated fats and is an essential factor influencing the differentiation fate of stem cells and the behaviors of osteoblast and osteoclast lineages.  The authors may also want to discuss the issues even if they were unable to display the changes in the expression cytokines/ adipokines in their results. 

Author Response

Dear Reviewer,

I appreciate your valuable comments! I have revised the paper as you suggested, which can be checked via Track Changes.

1. Line 95(p.3): it is believed that the rats used in the current study should be female; please correct the gender of the rats.

 Respond> I have revised that part as you suggested.

2. Lines 101-2(p.3): how do the authors ascertain that 8 weeks after the surgery is required to develop osteoporosis? If it is based on previous experiments, please indicate the reference.

 Respond> I have revised that part as you suggested.

3. Lines 222-4(p.6): it is difficult to tell if the authors use repeated measure ANOVA, a statistical method commonly used to analyze repeated measurements for humans or animals at different time points (such as the body weights of the rats in the current study). 

 Respond> I have revised that part.

4. Line 345(p.11, Figure 6): it should be panel "C" that displays the BMD measurements by DXA (rather than B). Please correct.

 Respond> I have revised that part as you suggested.

5. The authors intended to examine whether the overexpression of the enzyme SCD1 can modulate the loss of osteoblast differentiation induced by estrogen deficiency. Though the results showed that the authors did successfully mitigate the adiposity in the bone marrow and increase the osteogenic differentiation in their model, it is pitful that the results did not show how the overexpression of SCD1 enzymes affects the fatty acid composition of bone marrow.  It is known that fatty acids in bone marrow microenvironment also affect the differentiation and behaviors of both osteoblasts and osteoclasts.  The authors may, at least, include in their discussion how fatty acid composition in bone marrow may influence the differentiation and/or the fate of both bone cells.

 Respond> Thank you for your insightful comments. In light of your comments, we revised our discussion to include the impact of altered fatty acid composition on osteoblast and osteoclast differentiation and function.

6. Similarly, the expression of cytokines and adipokines also varies with the proportion of saturated fats and is an essential factor influencing the differentiation fate of stem cells and the behaviors of osteoblast and osteoclast lineages.  The authors may also want to discuss the issues even if they were unable to display the changes in the expression cytokines/ adipokines in their results. 

 Respond> Thank you for guiding me through this very important part. As you mentioned, cytokines and adipokines that can vary with the ratio of saturated fat will certainly affect cell metabolism. If possible, I will try to clarify the effects of these factors through follow-up studies to avoid the discussion being too broad in this paper. We appreciate you for suggesting the direction of the related research.

Thank you again for your valuable feedback; It helped me clarify the paper!

Sincerely,

Reviewer 2 Report

Comments and Suggestions for Authors

This is a nice, carefully conducted, original study investigating the effects of locally overexpressing Stearoyl-CoA Desaturase 1 (SCD1) (therefore of converting saturated into monounsaturated fatty acids) on osteogenic differentiation and bone regeneration in both in vivo and in vitro rat models

The material and methods are presented in great detail, the results are convincing and sound, proving that that saturated fatty acids inhibit osteoblast differentiation and by overexpressing the SCD1 gene to convert these saturated fatty acids into unsaturated fatty acids some effects are reverted.

Overall it is a very interesting study, paving the way for further reasearch aiming to elaborate novel therapeutic methods in osteoporosis

Author Response

Dear Reviewer,

We would like to thank you very much for your excellent review!

Sincerely,